# Subwords as Skills: Tokenization for Sparse-Reward Reinforcement Learning

**David Yunis**
TTI-Chicago
Chicago, IL
dyunis@ttic.edu

**Justin Jung**[*]
Springtail.ai
San Francisco, CA
justin@springtail.ai

**Falcon Z. Dai**[†]
Symbolica AI
San Francisco, CA
falcon@symbolica.ai

**Matthew R. Walter**
TTI-Chicago
Chicago, IL
mwalter@ttic.edu

## Abstract

Exploration in sparse-reward reinforcement learning (RL) is difficult due to the need for long, coordinated sequences of actions in order to achieve any reward. Skill learning, from demonstrations or interaction, is a promising approach to address this, but skill extraction and inference are expensive for current methods. We present a novel method to extract skills from demonstrations for use in sparse-reward RL, inspired by the popular Byte-Pair Encoding (BPE) algorithm in natural language processing. With these skills, we show strong performance in a variety of tasks, $1000\times$ acceleration for skill-extraction and $100\times$ acceleration for policy inference. Given the simplicity of our method, skills extracted from 1% of the demonstrations in one task can be transferred to a new loosely related task. We also note that such a method yields a finite set of interpretable behaviors. Our code is available at https://github.com/dyunis/subwords_as_skills.

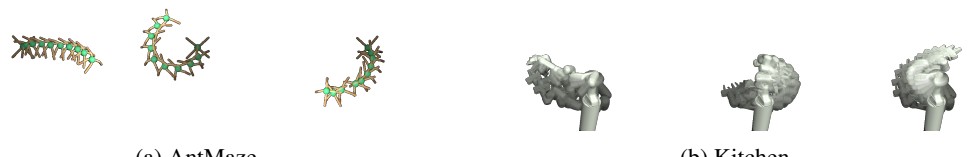

(a) AntMaze
(b) Kitchen

Figure 1: A sample of some "skills" that our method identifies for the (a) AntMaze and (b) Kitchen environments, where color is darker for poses earlier in the trajectory. Skills consist of linear motion and turning in AntMaze, and reaching and pulling motions in Kitchen. Our method discovers a finite inventory of skills, so it is possible to visualize and interpret them.

## 1 Introduction

The reinforcement learning (RL) paradigm, that allows an agent to interact with an a priori unknown environment and collect its own data, is a promising approach to learning in many domains where high-quality data collection is financially too expensive or otherwise intractable. Though it began with dynamic programming in tabular settings, the recent use of neural networks as function approximators has led to great success on many challenging learning tasks [47, 71, 26]. Typically, these successes owe to particular properties of the tasks. In some cases, it is simple to define a reward function that provides an informative learning signal at every step of interaction (the dense-reward setting),

---

[*]Work done while at University of Chicago.
[†]Work done while at Toyota Technological Institute at Chicago (TTI-Chicago).

38th Conference on Neural Information Processing Systems (NeurIPS 2024).

like directional velocity of a robot learning to walk [27]. In other cases, an environment model can aid search, as in the case of Chess or Go [71]. In all cases access to a fast simulator is paramount. However, for many natural tasks—like teaching a robot to make an omelet—it is much simpler to tell when the task is completed than to supervise each individual step or model the environment dynamics. Learning in these sparse-reward settings, where an informative reward is only obtained extremely infrequently (e.g., at the end of successful episodes), is notoriously difficult. In order for a learning agent to improve its policy, the agent needs to find reward, which requires long periods of exploration, often in a coordinated fashion. One solution to the sparse-reward problem is to engineer a proxy dense-reward, but that requires significant expertise and can lead to undesired reward-hacking behavior [73].

Another class of solutions to the exploration problem aims to create temporally extended actions, or "skills", from interaction [69, 22, 51, 52] or demonstrations [38, 72, 57, 2, 6, 46]. Formally, given a dataset of demonstrations $\mathcal{D} = \left\{ \left( s_0^{(i)}, a_0^{(i)} \right), \ldots, \left( s_t^{(i)}, a_t^{(i)} \right) \right\}_i$ of related behavior to the desired task, we want to extract a new action space $\mathcal{A}' \subset \cup_{i=1}^{\infty} \Pi_{u=1}^{t} \mathcal{A}_u$ consisting of sequences of the original action space, and then find a policy for the desired task using this action space, $\pi : \mathcal{S} \to \mathcal{A}'$. Current methods for skill-extraction rely on neural networks, which require large numbers of demonstrations and expensive training.

Like the long-range coordination required for exploration in sparse-reward RL, language models must model long-range dependencies between discrete tokens. Finer-grained character input leads to extremely long sequences and requires low-level modeling; coarser-grained word-level input results in the model poorly capturing rare and unseen words. The standard solution for language models is to create "subword" tokens somewhere in between individual characters and words, that can express any text [24, 67, 62, 39, 66, 31].

Lifting this idea from language modeling to RL, we propose a tokenization method for skill-learning from demonstrations: Subwords as Skills (SaS). Following prior work [18, 68], we discretize the action space where necessary and use a simple byte-pair encoding (BPE) scheme [24, 67] to obtain temporally extended actions. Then, we use this subword vocabulary as the action-space for online RL. As we demonstrate, such a method benefits from extremely fast skill-generation (seconds v.s. hours for neural network-based methods), $100\times$ faster rollouts due to the lack of an extra neural network during inference, and strong results in several sparse-reward domains. Additionally, we demonstrate transfer of skills collected in a different environment and we interpret the finite set of skills. Code is available for our experiments at https://github.com/dyunis/subwords_as_skills.

## 2   Related Work

**Exploration in RL:** Exploration is a fundamental problem in RL, particularly when reward is sparse. A common approach to encouraging exploratory behavior is to augment the (sparse) environment reward with a dense bonus term that biases toward exploration. This includes the use of (possibly approximate) state visitation counts [61, 45, 9, 13] and state entropy objectives [48, 30, 42, 60, 44, 79] that incentivize the agent to reach "novel" states. Related, "curiosity"-based bonuses encourage the agent to take actions in states where the effect is difficult to predict using a learned forward [65, 15, 74, 56, 1, 12] or inverse [29] dynamics model.

**Temporally Extended Actions and Hierarchical RL:** Another long line of work proposes action abstractions to enable more effective exploration [49] and simplify the credit assignment problem. Hierarchical reinforcement learning (HRL) [20, 34, 75, 11, 53, 54, 76, 21, 8, 40, 5, 77] considers the problem of learning policies with successively higher levels of abstraction, where the lowest level considers primitive actions in the environment and the higher levels reason over temporally extended transitions. A classic example of action abstractions is the options framework [76], which provides a standardization of HRL in which an option is a terminating sub-policy that maps states (or observations) to low-level actions. Options can be prescribed as predefined low-level controllers or learned via intermediate rewards [21, 20, 76]. Some simple instantiations of options include repeated actions [70] and self-avoiding random walks [3]. Konidaris and Barto [37] learn a two-level hierarchy by incrementally chaining options ("skills") backwards from the goal state to the start state. Nachum et al. [49] propose a hierarchical algorithm that learns in a sample-efficient, off-policy fashion. Such gains require addressing normal off-policy instability and non-stationarity that comes with jointly learning low- and high-level policies. Levy et al. [43] use different forms of hindsight [4]

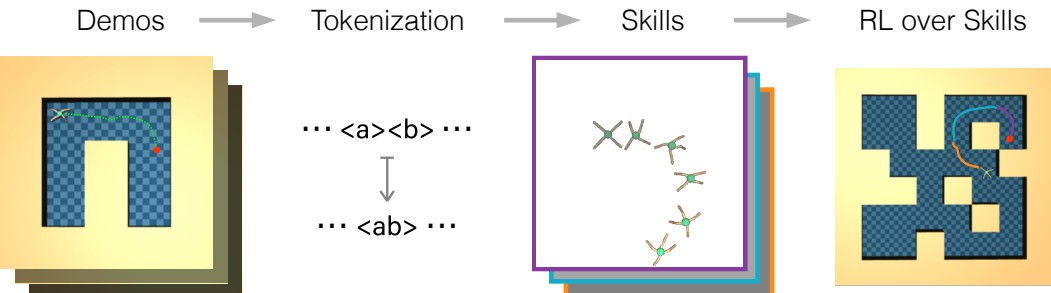

Figure 2: Abstract representation of our method. Given demonstrations in the same action space as our downstream task, we discretize the actions and apply a tokenization technique to recover "subwords" that form a vocabulary of skills. We then train a policy on top of these skills for a new task. We only require a common action space between demonstrations and the downstream task.

to address similar instability issues that arise when learning policies at multiple levels in parallel. One particularly related work applies grammar-learning to online RL [41], but such a method learns an ever-growing number of longer actions which is problematic in the sparse-reward setting.

**Skill Learning from Demonstrations:** In addition to the methods mentioned above in the context of HRL, there is an existing body of work that seeks to discover extended actions (skills) prior to, instead of during, online RL. Many methods have been developed for skill discovery from interaction [19, 25, 22, 78, 51, 52]. Most related to our setting is a line of work that explores skill discovery from demonstrations [38, 46, 2, 72, 57, 6]. As an example, Lynch et al. [46] learn a VAE [36, 64] on chunks of action sequences in order to generate a temporally extended action by sampling a single vector. Ajay et al. [2] follow a similar approach on top of entire trajectories and only rollout a partial trajectory at inference time. Some of these methods [2, 72, 57] condition on the observations when learning skills; however, such skills transfer poorly across domains unless they are trained on randomized environments [57, 6]. Others [46, 6] simply condition on actions, which means that the skills can be reused in any domain that shares the same action space. To extract more generalizable skills, we follow the latter approach. While a concurrent work [80] uses a method similar to ours in order to discover skills for supervised learning and transfer learning, we focus on the use of tokenization for online RL.

## 3 Method

Similar to prior work [46, 2, 72, 57, 6], we extract skills from demonstration data of action sequences. Formally, these sequences are a dataset of $N$ trajectories with lengths $\{n_i\}_{i=1}^N$ that involve the same action space as our downstream task:

$$\mathcal{D} = \left\{ (a_j)_i | i \in \{1, ..., N\}, \ j \in \{1, ..., n_i\}, \ a_j \in \mathcal{A} \subseteq \mathbb{R}^{d_{\text{act}}} \right\},$$

where $a_j$ denotes an individual action. After extracting skills from this dataset, we use these skills as the new action space for reinforcement learning on a downstream task. Crucially, our skills do not rely on observations in the demonstrations, which allows them to transfer to different environments even with very little data. In following subsections we detail our precise method.

### 3.1 Byte-Pair Encoding

Byte-pair encoding (BPE) was first proposed as a simple method to compress files [24], but it has recently been used to construct vocabularies for NLP tasks with a resolution in between characters and whole-words [66, 67, 39, 62, 31].

Given a long sequence of tokens (e.g., characters) and an initial fixed vocabulary, BPE consists of two core operations: (i) compute the most frequent pair of neighboring tokens and add it to the vocabulary, and (ii) merge all instances of the pair in the sequence. These two steps of adding tokens and merging alternate until a desired vocabulary size is reached.

## 3.2 Discretizing the Action Space

BPE requires an initial vocabulary $\mathcal{V}$ and data formatted as a string of discrete tokens. Clustering is a simple way to form discrete tokens from a continuous action space. Prior work has leveraged these ideas in similar contexts [32, 68, 33] and we follow suit. For simplicity, we perform $k$-means clustering with the Euclidean metric on the actions of demonstrations in $\mathcal{D}$ to form a vocabulary of $k$ discrete tokens $\mathcal{V} = \{v_0, \ldots, v_k\}$. Our default choice for $k$ will be two times the number of degrees-of-freedom (DoF) of the original action space, or $2 \times d_{\text{act}}$. We further study this choice in Appendix 4.5. Such clustering is the same as the action space of Shafiullah et al. [68] without the residual correction.

## 3.3 Merging and Pruning the Subwords

After discretizing the action space (if continuous), we can relabel our demonstrations so that trajectories consist of "strings" of action tokens. Then, we can run BPE [24] with a large final vocabulary size on these strings to extract skills.

As it runs, BPE keeps all intermediate subwords that make up the longest units. In the context of language, this redundancy may not be particularly detrimental. However, in reinforcement learning redundancy in the action space of a policy will result in a large number of similar actions that compete for probability mass, making exploration and optimization difficult. Thus, we prune the BPE vocabulary to a much smaller size.

To prune our vocabulary to a size $N_{\text{min}}$, we choose a desired maximum length of skills, say $L = 10$ actions long. For our pruned vocabulary, we take the first $N_{\text{min}}$ subwords of length $L$ that were found by BPE. If there are only $m < N_{\text{min}}$ subwords of length $L$ discovered, we then take the first $N_{\text{min}} - m$ subwords of length $L - 1$, and so on until we reach the desired vocabulary size of $N_{\text{min}}$. We choose the first subwords of a certain length because by the design of BPE, those will be the most frequent units of that length. If our demonstrations contain common and useful behavior, these will be the most frequent chunks. We provide an algorithmic description of our entire skill-extraction method in Algorithm 1.

Implicit in our method is an assumption that portions of the demonstrations can be recomposed to solve a new task, i.e., that there exists a policy that solves the new task with the action space that we choose. One can imagine a counter-example where the subwords we obtain lack some critical action sequence without which the new task cannot be solved, either because it is lacking in the demonstrations or because extraction is imperfect. Still, we will show that this assumption is reasonable for several sparse-reward tasks.

# 4 Experiments

In the following sections, we demonstrate the empirical performance of our proposed method: first extracting skills from demonstrations and then using those skills as an action space for online sparse-reward RL. Unlike common methods for offline RL, we do not use any information about observations or reward in the demonstrations. We see that our extracted skills provide significant speed benefits and sensible exploration behavior. We also compare our observation-free unconditional skills to observation-conditioned skills and discuss performance. We then examine the transfer setting, where demonstrations come from a different domain. Finally, we present an ablation of hyperparameters.

## 4.1 Reinforcement Learning with Unconditional Skills

**Tasks:** We consider online RL on AntMaze and Kitchen from D4RL [23], two very challenging sparse-reward state-based environments. AntMaze is a maze navigation environment with a quadrupedal robot where the reward is 0 except for at the goal, and Kitchen is a manipulation environment in a kitchen setting where reward is 0 except for on successful completion of a subtask. Demonstrations in AntMaze consist of trajectories between random start and end states in the same maze, while demonstrations in Kitchen consist of different sequences of subtasks than the eventual task. We also consider CoinRun [17], a discrete-action platforming game. Unlike AntMaze and Kitchen, CoinRun is a visual domain and the demonstrations are collected in levels distinct from those of the final task. All of these domains require many coordinated actions in sequence to achieve any reward, with

**Algorithm 1** Skill-extraction with BPE

---

1: Given action-only demonstrations $\mathcal{D} = \{(a_j)_i | i \in \{1, ..., N\}, \ j \in \{1, ..., n_i\}, \ a_j \in \mathcal{A} \subseteq \mathbb{R}^{d_{\text{act}}}\}$

2: Given number of clusters $k$, max vocab size $N_{\text{max}}$, skill length $L$, desired vocab size $N_{\text{min}}$

3:

4: **if** the action space is not discrete, i.e. $\forall j, \ a_j \notin \mathbb{N}$ **then**

5:  Run $k$-means on actions with $k$ clusters to get discrete actions $\mathcal{V} = \{v_i\}_{i=1}^k$

6:  Replace $a_j$ in $\mathcal{D}$ with closest discrete action index, $a_j \leftarrow \text{argmin}_{1 \le l \le k} \|a_j - v_l\|_2$

7: **else**

8:  Use the discrete action space as seed vocabulary $\mathcal{V} \leftarrow \mathcal{A}$

9: **end if**

10:

11: Initialize subword vocabulary $\mathcal{W} = \mathcal{V}$

12: **while** $|\mathcal{W}| < N_{\text{max}}$ **do**

13:  Find most common pair of neighboring subwords $w_i, w_j \in \mathcal{W}$ in demonstrations $\mathcal{D}$

14:  Merge pair into a new subword, $w' = \text{concat}(w_i, w_j)$, and add to vocabulary, $\mathcal{W} \leftarrow \mathcal{W} \cup \{w'\}$

15:  Relabel demonstrations $\mathcal{D}$, replacing sequences of $(w_i, w_j)$ with $w'$

16: **end while**

17:

18: Initialize final vocabulary $\mathcal{W}' = \varnothing$

19: **while** $|\mathcal{W}'| < N_{\text{min}}$ **do**

20:  $n \leftarrow$ number of subwords $w \in \mathcal{W}$ with length $L$

21:  $\mathcal{W}' \leftarrow \mathcal{W}' \cup \{\text{first } \min(n, N_{\text{min}} - |\mathcal{W}'|) \text{ subwords of length } L \text{ that were merged}\}$

22:  $L \leftarrow L - 1$

23: **end while**

24:

25: **return** $\mathcal{W}'$

---

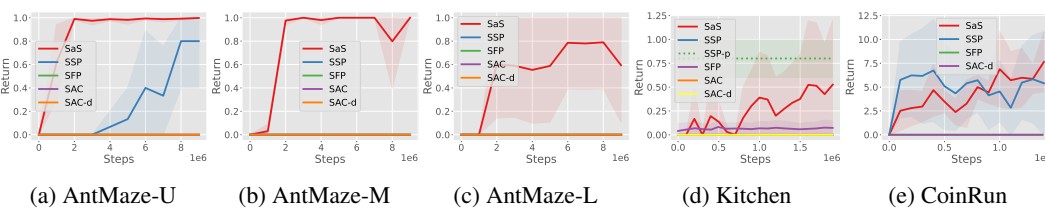

|   (a) AntMaze-U   |   (b) AntMaze-M   |   (c) AntMaze-L   |   (d) Kitchen   |   (e) CoinRun   |

Figure 3: Main comparison (unnormalized scores). SSP corresponds to results from official code of Pertsch et al. [57], SSP-p corresponds to published results. AntMaze is scored 0–1, Kitchen is scored 0–4 in increments of 1, CoinRun is scored 0–100 in increments of 10. CoinRun is a discrete-action domain, so instead of SAC only SAC-discrete can be used. We see strong performance when compared to baselines across tasks.

horizons between 280 and 1000 steps. See Appendix A for more information on the tasks and data. Due to the suboptimality of demonstrations on AntMaze, we filter demonstrations to remove portions that correspond to jittering in place.

**Baselines:** We consider SAC [28]; SAC-discrete [16] on top of our discretized $k$-means actions; Skill-Space Policy (SSP), a VAE [36, 64] trained on sequences of 10 actions at a time [57]; and State-Free Priors (SFP) [6], a sequence model of actions that is used to inform action-selection during SAC inference. For SAC we use a standard implementation. For SAC-discrete we reimplement the method. For SSP we use the official implementation [59] and tune hyperparameters for new domains. For SFP we use official code [7], and are unable to tune hyperparameters due very large runtimes. Figure 3 provides the complete set of results. We report mean and standard deviation across five seeds. As defaults for our method, we use $k = 2 \times d_{\text{act}}$ and $N_{\text{min}} = 16$. We choose $N_{\text{max}} = 10^6$ so that we always find sufficiently many skills with the desired length $L = 10$, which we choose to be comparable with SSP's length 10. We ablate these choices in Appendix 4.5. For more experimental details and hyperparameter settings, see Appendix A. Including our method, all methods only use

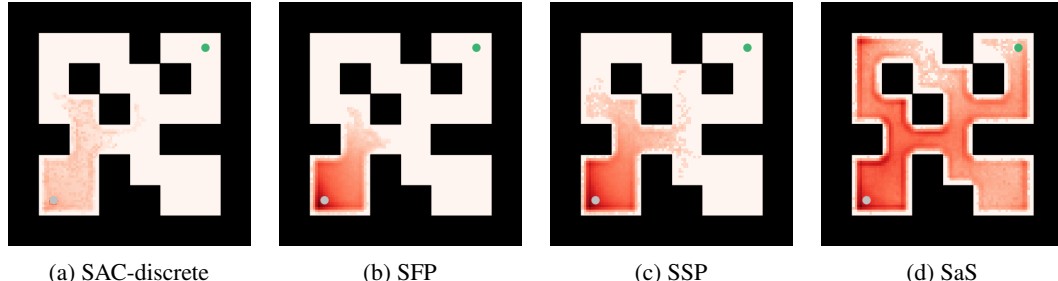

| (a) SAC-discrete | (b) SFP | (c) SSP | (d) SaS |

Figure 4: A visualization of state visitation in online RL on AntMaze Medium in the first 1 million timesteps for (a) SAC-discrete, (b) SFP, (c) SSP, and (d) our method averaged over 5 seeds. The grey circle in the bottom-left denotes the start position, while the green circle in the top-right indicates the goal. Notice that our method explores the maze much more extensively, with exploration behavior that is similar for all five seeds. SAC's visitation is tightly concentrated on the start state, which is why there is so little red in (a) the visitation rendering for SAC-discrete (i.e., it is occluded by the gray circle).

the action sequences of demonstrations. For AntMaze, we take the best setting of SSP whether the demonstrations are filtered or not.

We see in Figure 3, that even in these challenging sparse-reward tasks, our method can perform well. We show strong performance over baselines, which mostly achieve 0 return, except for in CoinRun where we are competitive. The large standard deviations are due to the fact that we can only run a small number of seeds and some seeds fail to achieve any reward, but we will show that exploration behavior is still reasonable, which gives us more confidence in the conclusions.

Due to the simplicity of our method, it is significantly faster than baselines. In Table 1, we measure the wall-clock time required to generate skills, as well for a single rollout. We see that our method achieves extremely significant speedups compared to prior work. Our skill discovery method is fast as we simply need to run $k$-means and tokenization. SSP and SFP require training larger generative models. In the case of rollouts, our method predicts an entire sequence of actions using a simple policy every $L$ steps, while SSP and SFP require larger models in order to predict the latent variable, and then generate the next action from that latent. The speedup of our method also translates to faster RL (around 10 hours for our method vs. 24 hours for SSP and 1 week for SFP).

Table 1: Timing (mean $\pm$ one standard deviation) on AntMaze Medium in seconds. Methods measured on the same Nvidia RTX 3090 GPU with 8 Intel Core i7-9700 3 GHz CPUs @ 3.00 GHz. SSP takes $\sim$36 hours for skill generation and SFP takes $\sim$2 hours.

| Method | Skill Generation | Online Rollout |
|---|---|---|
| SSP | $130000\pm_{1800}$ | $0.9\pm_{0.05}$ |
| SFP | $8000\pm_{500}$ | $4.1\pm_{0.1}$ |
| SaS | $\mathbf{3\pm_1}$ | $\mathbf{0.007}\pm_{\mathbf{0.0006}}$ |

### 4.2 Exploration Behavior on AntMaze Medium

The stringent evaluation procedure for sparse-reward RL equally penalizes poor learning and exploration. In order to shed light on the poor performance of some methods in Figure 3, we examine exploration on AntMaze Medium. We choose this domain because it is straightforward to visualize good and bad exploration behavior by plotting maze coverage. In Figure 4 we plot state visitation for the first 1 million of 10 million steps of RL. We show the approximate start position in grey in the bottom left and the approximate goal location in green in the top right. Higher color saturation corresponds to a higher probability of that state. Color is scaled nonlinearly according to a power law between 0 and 1 for illustration purposes. Thin white areas between the density and the walls can be attributed to the fact that we plot the center body position, and the legs have a nontrivial size limiting the proximity to the wall.

Figure 4 visualizes the exploration behavior across methods, averaged over 5 seeds. We see that the 0 values for return in Figure 3 for SAC, SSP and SFP are likely due not to poor optimization, but rather to poor exploration early in training, unlike our method. Indeed, we show in Appendix C that

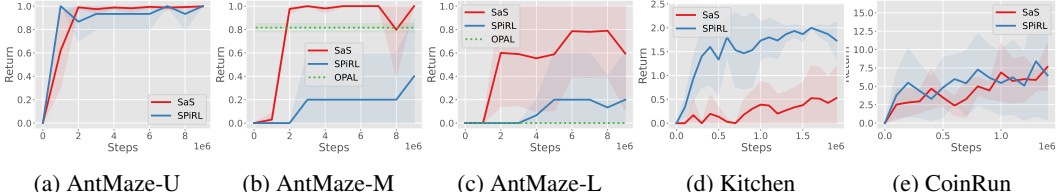

| (a) AntMaze-U | (b) AntMaze-M | (c) AntMaze-L | (d) Kitchen | (e) CoinRun |

Figure 5: Comparison to methods with observation-conditioned skills. In general we see conditioning helps when the data closely overlaps with the downstream task (Kitchen), but not in AntMaze where the demonstrations are somewhat disjoint. OPAL is a closed-source method similar to SPiRL, so results are taken from Ajay et al. [2, Section 5.3].

on AntMaze Large, for which not all seeds succeeds (unlike AntMaze Medium, for which all seeds succeed), seeds that perform poorly still exhibit good exploration behavior. One reason for this could be due to the fact that our subwords are a discrete set, so the policy always has diverse options to pick, whereas continuous latent variables can model infinitely many skills with only minor differences. In addition, SAC has fundamental issues in sparse-reward environments as the signal to the Q-function is driven entirely by the entropy bonus, which will lead to uniform weighting on every action and as a result, Brownian motion in the action space. Such behavior is likely why the default setting for SAC [28] aggressively drives the policy to determinism, but in the sparse reward setting, this also results in a uniform policy. Without diverse and long sequences of coordinated actions, such uniform exploration is insufficient.

### 4.3 Comparison to Observation-Conditioned Skills

Our skill extraction method does not rely on observations and so may lead to more generalizable skills. However, not conditioning on the observations comes with the drawback that a policy needs to learn the context to deploy skills from scratch. Alternatively, observation-conditioned skills bias policy exploration to match that of the demonstrations. This allows for more stable exploration [57, 58, 2], but worse generalization [6].

**Baselines:** Here we compare to the observation-conditioned extension of SSP, SPiRL [57] which biases a policy toward the use of skills in the same context as in the demonstrations. We also include OPAL [2], a concurrent work with SPiRL. We take numbers from the paper as OPAL is closed-source. Our tuning procedure for SPiRL is similar to SSP, where we consider the best setting over filtered and unfiltered demonstrations.

In Figure 5, we see that SPiRL shows very strong performance on Kitchen, where the overlap between the dataset and the downstream task is exact, but struggles with AntMaze, likely due to differences between the random trajectories in the dataset and the final task. We also note that our result for SPiRL in Kitchen is worse than the reported 2–3 [57]. Given that we use the official code, which already implements Kitchen, the difficulty of sparse-reward RL is likely to blame.

### 4.4 Transferring Skills

One benefit of unconditioned skills is that they can be extracted from demonstrations that differ from the final task domain. In Figure 6, we highlight that such transfer is possible, and that with varying percentages of demonstrations (down to 10 trajectories) performance is fairly stable. It may seem odd that 1% performs better than 10% and 25%, but this may be explained by the bias that random subsampling imposes on the demonstrations. By contrast, observation-conditioned methods require large amounts of trajectories in randomized environments to transfer effectively [57, 6].

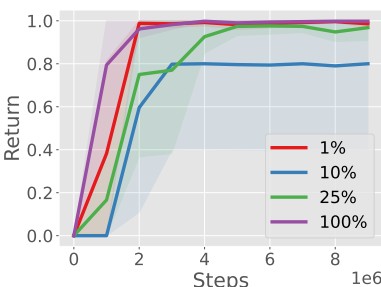

Figure 6: Results on transferring skills extracted from AntMaze-M to downstream RL on AntMaze-U, with varying quantity of demonstrations. Even with 1% of the data, our method extracts useful skills

## 4.5 Ablations

There are a few key hyperparameters of our method ($k$, $N_{\min}$ and $L$). In the following, we perform ablations over them in the AntMaze Medium and Kitchen environments. In general, behavior in the Kitchen environment is much noisier, which may indicate that RL training is still unstable.

**Number of Discrete Primitives** All of our results in Figure 3 use the simple rule-of-thumb that $k = 2 \times$ degrees-of-freedom. In Figure 7 we see that this choice seems to be acceptable, though it should be noted that significantly larger values of $k$ lead to shorter skills as there are fewer and fewer common subwords with the desired length $L$.

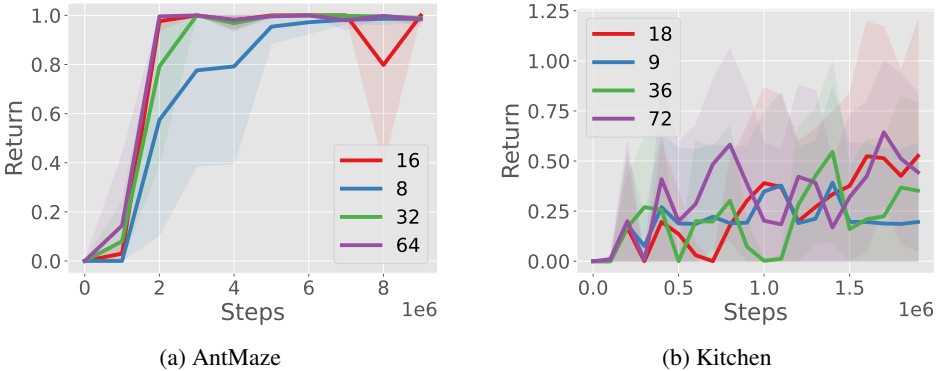

(a) AntMaze  (b) Kitchen

Figure 7: Results for different numbers of clusters. For AntMaze, DoF = $d_{\text{act}} = 8$, Kitchen DoF = $d_{\text{act}} = 9$, and the default setting is $k = 2 \times d_{\text{act}}$. Note the legend is left unsorted so that the default setting $k = 2 \times d_{\text{act}}$ is rendered in a consistent color and position across all plots.

**Subword Length** A crucial property of the vocabulary is the length of the subwords. Long subwords lead to more temporal abstraction and easier credit-assignment for the policy, but long subwords can also get stuck for many transitions, possibly leading to poor exploration. In Figure 8, we vary the value of subword length $L$. Our default setting for each environment uses $L = 10$ to match the baselines, but we see that different values are also acceptable, though $L = 5$ makes RL more difficult in AntMaze.

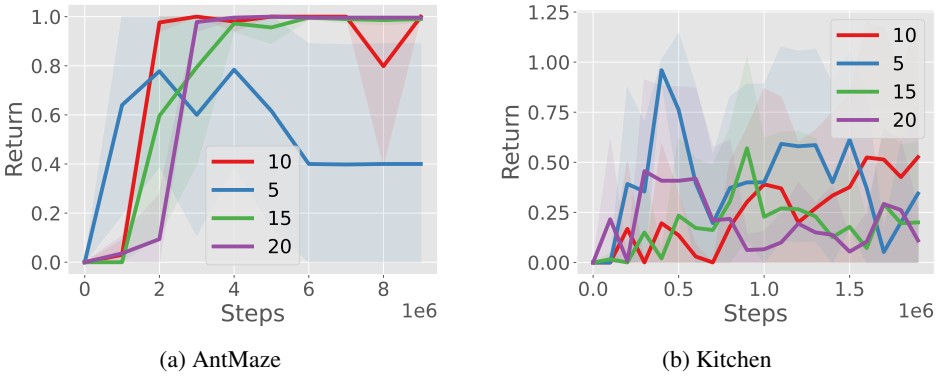

(a) AntMaze  (b) Kitchen

Figure 8: Results for different choices of subword length $L$, where the default setting is $L = 10$. Note the legend is left unsorted so that the default setting $L = 10$ is in a consistent color and position.

**Vocabulary Size** Ultimately, the dimensionality of the action space will make exploration easier or harder. A large vocabulary results in too many paths for the policy to explore well, but a vocabulary that is too small may not include all the skills necessary to represent a good policy for the task. We see in Figure 9 that larger vocabulary sizes do in fact make RL more difficult in AntMaze.

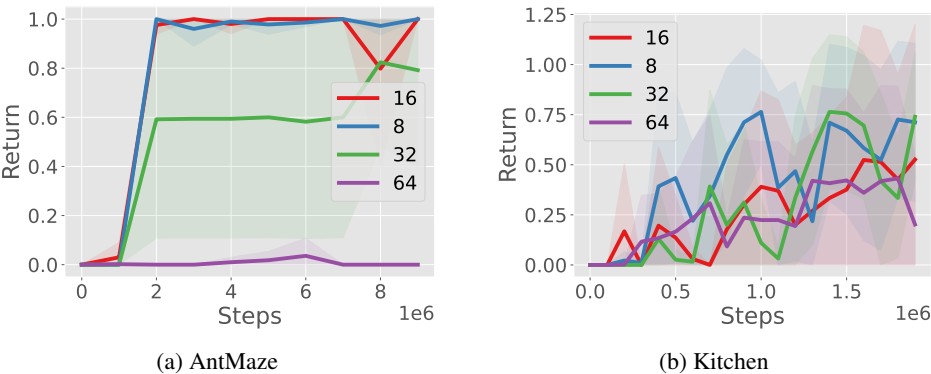

(a) AntMaze

(b) Kitchen

Figure 9: Results for different choices of vocabulary size $N_{\min}$, where the default setting is $N_{\min} = 16$. Note the legend is left unsorted so the default setting $N_{\min} = 16$ is a consistent color and position.

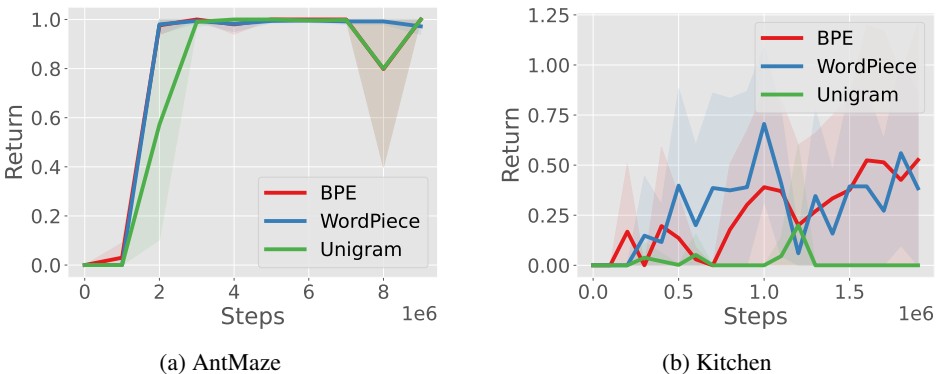

(a) AntMaze

(b) Kitchen

Figure 10: Results for different choices of tokenizer algorithm, where BPE is the default.

**Tokenizer Algorithm**   All of the results thus far have only considered the BPE tokenizer [24], but other tokenizers have seen benefits in language modeling, like WordPiece [67] or Unigram [39]. We see in Figure 10 that BPE and WordPiece are somewhat interchangeable, but that performance suffers with Unigram. This is likely because, unlike BPE and WordPiece, Unigram does not discover a hierarchically structured vocabulary where shorter subwords are contained in longer subwords. Thus, naively picking the first $N_{\min}$ subwords of length $L = 10$ may not extract the most common behavior. If we were to allow a length-independent vocabulary, Unigram might be a more natural choice, but we did not explore that here due to the necessity of comparing fairly with baselines.

## 5   Conclusion

Architectures from NLP have made their way into offline RL [14, 32, 68], but as we have demonstrated, there is a trove of further techniques to explore. Motivated by prior evidence that the full range of the action space is not required, we discretize and form skills through a simple tokenization method. Our method is much faster in skill generation and policy inference and leads to strong performance in several challenging sparse-reward tasks with a relatively small sample budget. In addition, the finite vocabulary size lends itself to interpretable skills: one can simply look at the execution to figure out what has been extracted (Appendix B). As proposed, however, there are a few key limitations. Discretization removes resolution from the action space, which may be detrimental in settings like fast locomotion (Appendix D), but this may be fixed by using more clusters $k$ or a residual correction [68]. In addition, like prior work execution of our subwords is open-loop, so exploration may be inefficient [3] and unsafe [50]. Still, given the speed, performance and interpretability advantages, we believe that our tokenization method is the first step on a new road to efficient reinforcement learning.

## Acknowledgments

We thank Takuma Yoneda and Jiading Fang as well as other members of the RIPL lab at TTIC for helpful discussions throughout the process. This material is based upon work supported by the National Science Foundation Graduate Research Fellowship Program under Grant No. 1754881. Any opinions, findings, and conclusions or recommendations expressed in this material are those of the authors and do not necessarily reflect the views of the National Science Foundation.

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

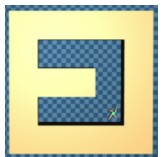 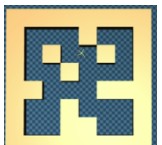 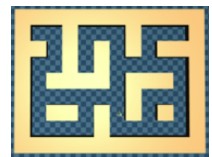 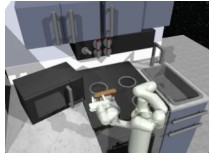 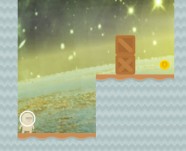

(a) AntMaze Umaze (b) AntMaze Medium (c) AntMaze Large      (d) Kitchen      (e) CoinRun

Figure 11: Offline environments, figures courtesy of Fu et al. [23] and Cobbe et al. [17]. For AntMaze Umaze the starting location is in the bottom left, and the goal is in the top left. For AntMaze Medium and Large the starting locations are in the bottom left, and goals are in the top right.

## A    Online RL Experimental Details

### A.1    Data

As a set of diverse and challenging sparse-reward tasks, we select AntMaze and Kitchen from D4RL [23] and CoinRun [17]. We choose AntMaze as a much more challenging version of the PointMaze task considered in prior work [57, 6], and Kitchen as the most complicated manipulation environment considered by Pertsch et al. [57]. CoinRun is chosen as a challenging discrete-action environment.

**AntMaze** An environment in which a MuJoCo Ant robot is tasked with navigating a maze (Figures 11(a), 11(b) and 11(c)). The observation space consists of positions and joint angles of the body geometries, while actions correspond to joint torques. Crucially, no information about the maze layout is given, so the agent must learn this through exploration. Reward is $0$ unless within a small distance $\epsilon$ of the goal, in which case it is $1$. Demonstrations from the dataset consist of a non-RL agent navigating between random start and end points within the maze [23]. In particular, the demonstrations are highly suboptimal, often crashing into walls, flipping over, and getting stuck. In order to extract nontrivial behavior with our method, we filter this data so that 10-step chunks that fail to move past a certain threshold normalized distance in observation space are dropped. We consider the best setting of filtered or unfiltered data for baselines.

**Kitchen** An environment in which a Franka Panda arm is tasked with performing a sequence of $4$ subtasks in a mock kitchen environment (Figure 11(d)). Example subtasks include moving a kettle between burners, turning on the stove, and opening the microwave. Observations consist of position and joint angles of the arm, as well as positions of key objects to be manipulated, and actions are joint torques. Once again, no information about the layout is given to the agent, and instead, it must be learned through exploration. Rewards are $0$ unless the correct subtask is completed in the correct order, which yields a reward of $1$. There are $4$ subtasks to be completed, so there is a maximum reward of $4$ available. Demonstrations are collected by humans using a VR interface [23], and consist of near-perfect executions of different sequences of $4$ subtasks from the final sequence.

**CoinRun** A procedurally-generated platforming game that involves traversing obstacles and avoiding enemies in order to reach a final goal (Figure 11(e)). Each level has a different layout and visual style, designed by humans, in order to require more general recognition from the policy. Observations consist of a $64 \times 64$ visual observation of the scene, centered on the agent, with velocity information painted into the upper-left corner. Actions are discrete and consist of moving, jumping, and staying still. Reward is $0$ until the final goal for a level is reached, in which case it is $10$. For RL, we select a fixed subset of $10$ "hard" levels in sequence for an agent to complete, to mimic classic games, so the maximum possible reward is $100$. We collect demonstration data through playing around $100$ "easy" levels with different layouts and visual style than the eventual levels we perform RL on.

### A.2    Model

For the model, we choose a 4-layer MLP with $256$ hidden units in each layer. We use the default initialization in Stable Baselines 3 [63].

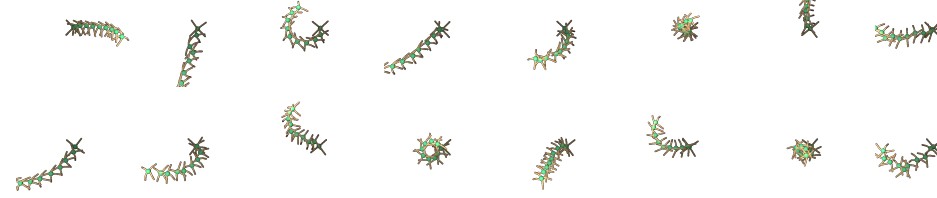

Figure 12: All skills discovered for AntMaze-M where color is darker for poses earlier in the trajectory. We see a range of linear motion and turning behaviors.

### A.3 Optimization

For our RL agent, we use SAC-discrete [16]. Both critics as well as the policy are optimized with Adam [35] with a standard learning rate of $3e - 4$. Replay buffer size is set to the standard $1$ million transitions. We update both critics and the policy every step of environment interaction and sample uniformly from the replay buffer to do so. Unlike Christodoulou [16], we follow a similar convention to Haarnoja et al. [28] and automatically optimize $\alpha$. We choose a target entropy dependent on the domain: $0.1$ for AntMazes, $0$ for Kitchen, and $0.5$ for CoinRun, though we find this hyperparameter to be relatively unimportant. More importantly, we found a large batch size crucial to good performance in AntMaze, where we use a batch size of $4096$. For other tasks, we use a batch size of $64$. Other hyperparameters are kept to their default values following SPiRL. Baselines are kept with original hyperparmeters for the domains they studied, and given it was the critical hyperparameter for our method, we tune batch sizes for SSP and SPiRL, but find default hyperparameters perform best. Because SFP is so expensive to run, we do not have the computational budget to tune hyperparameters.

For AntMaze we train for $10$ million steps, for Kitchen we train for $2$ million, and for CoinRun we train for $1.5$ million steps. All numbers come from $5$ random seeds.

### A.4 Skill-extraction hyperparameters

For AntMazes and Kitchen, we choose defaults of $k = 2 \times d_{\text{act}}$, $L = 10$, $N_{\max} = 10^6$ and $N_{\min} = 16$. For CoinRun there is no need for discretization, so we only choose $N_{\max} = 10^6$, $L = 10$, and $N_{\min} = 16$. These defaults are chosen to match the length $10$ skills of SSP.

### A.5 Implementation

Code was implemented in Python using PyTorch [55] for deep learning, Stable Baselines 3 [63] for RL, and Weights & Biases [10] for logging. It is available at https://github.com/dyunis/subwords_as_skills.

### A.6 Computational Requirements

All experiments were performed on an internal cluster with access to around $100$ Nvidia 2080 Ti (or more capable) GPUs. Each single run fits in around $2\,\text{GB}$ of GPU memory on a single machine. On AntMaze, training for our method typically takes around $10$ hours for a single run, while SSP [57] takes $24$ hours and SFP [6] takes over a week. In particular, this highlights exactly how poor the scaling can be for methods that call a large model at every transition.

## B   Qualitative Description of Skills

One nice property of our method is that, given that we extract a finite vocabulary, we can inspect the discovered skills. Below, we discuss the AntMaze and Kitchen domains as an example. In order to visualize skills, we execute the subwords for $100$ steps in the environment, and visualize the resulting trajectory. The actual duration of a skill is much shorter, but this is done to make the motions very clear.

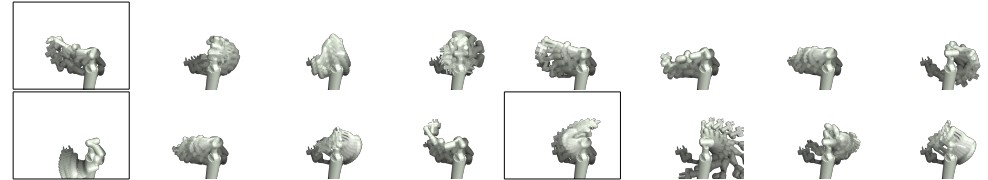

Figure 13: All skills discovered for Kitchen where color is darker for poses earlier in the trajectory. We see a range of different behaviors across the skills, including reaching (top row, first column), pulling (bottom row, first column), and pushing (bottom row, fifth column) motions.

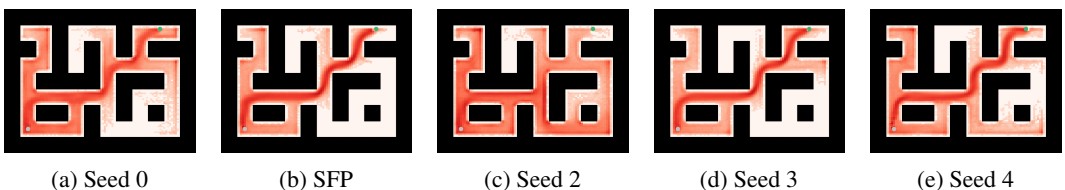

| (a) Seed 0 | (b) SFP | (c) Seed 2 | (d) Seed 3 | (e) Seed 4 |

Figure 14: A visualization of state visitation in online RL on AntMaze Large for all 10 million timesteps across different seeds. The grey circle in the bottom-left denotes the start position, while the green circle in the top-right indicates the goal. Notice that our method explores the maze much more extensively, with exploration behavior that is similar for all five seeds, even if the performance of some of those seeds is 0 in the eventual evaluation. In particular, Seeds 0 and 2 do not result in good evaluation performance regardless of the sensible exploration behavior.

In Figure 12, we see the skills extracted in AntMaze. In particular, we see turning in both directions, with differing turn radii, as well as various linear motions. It is straightforward to imagine why one would need both in designing an action space, and it seems that there are few explicit repetitions (though many variations on the theme).

In Figure 13, we visualize the different skills discovered in the Kitchen domain. These are difficult to present in a static form, as it is not simple to visualize interaction with the environment, but they consist of a variety of reaching and rotational motions that are useful for interacting with different objects. In the bordered images, we highlight three particular skills. In the top left is a reaching skill that might be used for reaching the light switch/oven knobs. In the bottom left is a pulling skill that could be useful opening a door. Lastly there is a pushing skill, that might be useful for sliding a door.

## C    Exploration Behavior when RL fails

Figure 4 visualizes the exploration behavior of our method on AntMaze Medium, for which all five seeds of our method succeed. We believe that it is informative to consider the exploration behavior on domains for which some seeds fail. To that end, we analyze the differences in exploration behavior of SaS on Antmaze Large, for which not all seeds achieve perfect performance. We see in Figure 14 that even those seeds that do not perform well have quite sensible exploration behavior and good coverage of the maze, so it seems like the major issue has to do with optimization in the RL setting, not exploration.

## D    Effect of Discretization In Locomotion

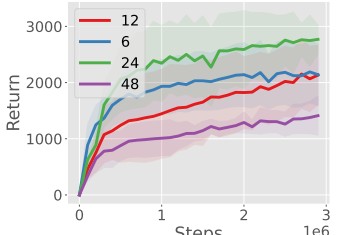

As mentioned in Section 5, discretization may remove resolution from the action space that could be useful, in particular for fine-grained manipulation or fast-locomotion tasks. To study this limitation, we investigate the effect of varying the discretization level on the Hopper locomotion environment from D4RL [23]. We use hyperparameters $k = 12$, $N_{\min} = 16$, $L = 5$ and train 5 seeds for 3 million steps each.

Figure 15: Experiments on the Hopper domain for varying number of clusters $k$.

In Figure 15, we see that the conclusions are curious. Finer discretization helps up to a certain point, after which it hurts performance. We are not totally certain as to why this happens. One hypothesis is that finer levels of discretization naturally results in shorter skills, as there are fewer repeated subwords, but this might make RL in a dense reward environment easier, not harder. In any case, all runs are worse than training on continuous actions.

## E   Effect of Data Quality

To see how our method performs with different kinds of data quality, we again use the Hopper environment from D4RL [23]. This is because, unlike sparse-reward tasks considered in the rest of the paper, Hopper provides a clear delineation of demonstration quality: "Random" for transitions from a random policy, "Medium" for transitions from a policy partway through training, and "Expert" for transitions from a policy at the end of training. We set $k = 12, N_{\min} = 16, L = 5$ and train 5 seeds for 3 million steps.

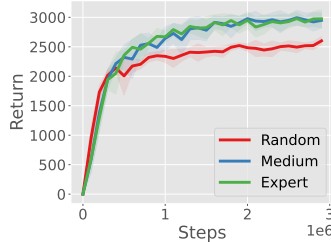

Figure 16: Experiments with demonstration data of varying quality in the Hopper domain.

From Figure 16, "Expert" demonstrations provide the best skills, "Medium" demonstrations are very similar to "Expert", but surprisingly "Random" demonstrations are quite competitive. What is likely happening here is that random demonstrations do not have enough common subwords, so the discovered subwords are quite short in length. Thus, with a dense reward it is still possible to recover good behavior.

