# OpenReview forum: "Subwords as Skills: Tokenization for Sparse-Reward Reinforcement Learning"
_NeurIPS.cc/2024/Conference — NeurIPS 2024 poster_

### Official Review · Reviewer_jRfS · 2024-06-14

**Soundness:** 4
**Presentation:** 3
**Contribution:** 3
**Rating:** 7
**Confidence:** 5

**Summary:**

This paper presents a method which discretizes action trajectories from offline data into skills through a BPE-inspired tokenization method. These discrete action trajectory skills are given directly to a high-level agent to utilize for solving downstream tasks.

**Strengths:**

**Presentation:** Writing is clear and the method is intuitive and simple, the paper is overall easy to understand. The main figure also does a good job of distilling the algorithm into a simple illustration.

**Method:** The method is extremely compute-efficient in skill creation. It’s also faster when using for downstream RL because the action sequences are unique skills, no forward passes are needed.

**Results:** Overall, results are convincing in that the method works on par with or better than other non-state conditioned skill-based RL methods.

**Experiments:** The authors performed a good amount of ablations and the experiments are performed across a few locomotion, manipulation, and discrete procgen settings that seem pretty comprehensive for an RL paper.

**Weaknesses:**

**Method:**

- One limitation of this method is that it *cannot be state conditioned*. Figure 5 clearly demonstrates that state-conditioned methods that can utilize state-based priors (e.g., SPiRL) can perform much better depending on the problem setting (Franka Kitchen, where there is a lot of overlap between the pre-training data and testing environment). On the other hand, methods like SPiRL can also be used without state-conditioning. This thus is less flexible in this regard.
- Another limitation that stems from the comparison in the introduction against large language models and NLP is that in NLP, associations between text tokens are learned during pre-training time and then utilized for downstream tasks. The authors’ method does not utilize this intuition despite being influenced by NLP. Instead, the policy must learn how to associate the action sequence tokens from scratch during RL training time, possibly wasting many environment steps just learning the associations. One way to fix this would be to learn some state-conditioned associations between skills/some prior over how to associate discrete action sequences together.

**Questions:**

- Why choose a specific desired length $L=10$ for the action sequences? A more flexible method would be able to use skills of varying lengths to better solve the task like https://openreview.net/pdf?id=r4XxtrIo1m9 or the already cited CompILE work by Kipf et al. It seems like the proposed method could easily be adapted for variable length skills too.
- Why are the ablations performed on Hopper (appendix D + E) but no Hopper experiments are in the main paper?
- Because this came out ~3 months before neurips submissions, it seems reasonable the authors did not cite it yet. But, this paper https://arxiv.org/abs/2402.10450 is extremely similar in the use of BPE for creating skills to use for downstream policy learning. How does your method compare to this?

As such, when using the NeurIPS rating scale, I'm currently giving this paper a 6 because I believe it is of moderate impact (results are adequate, method is nice and simple, but I'm not sure it would be considered high-impact, i.e. a 7, and there are some flaws and remaining questions).

**Limitations:**

Sufficiently addressed

---

> ### Author Rebuttal · Authors · 2024-08-07
>
> Thank you for your care in reading our paper. We appreciate the positive feedback on the clarity, and the comment on efficiency and the strength of the results, including a “pretty comprehensive” evaluation. We respond to your concerns below.
>
> 1. Method cannot be state-conditioned
>
> Though it is true that we present an unconditional method, it is completely possible to adapt into a state-conditioned method. After discovering skills and tokenizing the demonstrations via BPE, one can take all the observation-action pairs that correspond to, say, 3-action prefixes of our skills, and train a classifier to map from observations to one of these prefixes. Then, one can bias the sampling during RL toward skills whose prefix is more likely under the prior, in exactly the same way that SPiRL does. We did attempt to prototype this method during the rebuttal, but it was not possible to properly tune and compare to prior work due to time constraints. We do consider it interesting for future work.
>
> 2. Policy must learn association.
>
> This is intentional. Our goal here is to drive strong exploration, even when the collected demonstration data is small or out of domain (Figure 6). Having a strong skill prior requires lots of in-domain data collected, which is the language modeling case. That being said, we agree that learning a prior is a reasonable solution in such settings. See above for how we might go about doing so.
>
> 3. Why choose a specific skill length?
>
> This choice was made to make comparisons to baselines fair. We agree that BPE is perfectly suited to discover skills of different lengths which would be more flexible and makes more sense.
>
> 4. Why are ablations on Hopper?
>
> One of these ablations explores the effect of discretization in a task where we might expect it to matter more as hopper only has a single contact point. The sparse-reward tasks explored in the main text are quite resilient to discretization (Ant and Franka are stable) so it made less sense to do this for those tasks. For the ablation on data quality, D4RL does not provide quality splits for the Ant and Franka tasks, and we could try to synthesize quality splits by combining with random data, but this seemed like a poor proxy. Hopper provides these splits.
>
> 5. Related work?
>
> Thank you for pointing out this very related work and concurrent work, we will be sure to add a citation! It appears they have a similar use of BPE, but their goals are different: they discover BPE skills in the offline setting and then test the ability to generalize to downstream tasks with additional finetuning, still in the offline setting. Nowhere do they concern themselves with exploration or the online setting, and they use the entire vocabulary without pruning, which would be impossible for us as we show.

---

> > ### Comment · Reviewer_jRfS · 2024-08-09
> >
> > Thanks for the rebuttal, my questions are answered and I'll be raising my score to a 7.

---

> > > ### Author Response · Authors · 2024-08-11
> > >
> > > We are glad that our response was able to answer your questions and change your opinion of the paper for the better. Thank you for the engagement and careful read.

---

### Official Review · Reviewer_g8NK · 2024-07-10

**Soundness:** 3
**Presentation:** 2
**Contribution:** 3
**Rating:** 7
**Confidence:** 2

**Summary:**

This paper uses Byte pair encoding to create a discretised action space for RL from demonstrations. The authors show that Byte pair encoding can:
* improve exploration in sparse reward settings
* creating the skill action space is computationally cheap compared to methods that train deep learning models
* their approach is not conditioned on observation, which improves generalisation

**Strengths:**

The authors present a clear explanation of how (as far as I know) novel method for extracting skills from demonstrations. What is really cool is that you only to run k-means clustering, making this approach scale favorably to other approaches that use deep learning.

Also give the success of BPE in NLP, this is a very intuitive idea that seems like a great contribution to the RL community.

The ablations answered most of my questions about the design of their method.

**Weaknesses:**

It is difficult for me to understand the baselines and some details of the experiments. For example, its not clear how many demonstrations are used for each of the methods. Presumably as it only using k-means and BPE, it can work with a small number of demonstrations (this is hinted at on the skill transfer results). I assume for example, that offline RL methods cannot be used because the dataset of demonstrations is too small? It would be good to explain these basics to someone (like me) who is not familiar with this subfield.

Also, from the paper alone, I cannot tell whether the baselines are the state of the art approach for skill extraction.

To help with these weaknesses I would like:

* details on the number of demonstrations used
* more explanation of what the baselines are and how they work

For the transferring skills, it would be good to run the baseline (even if it is very poor) to plot in figure 6.

**Questions:**

How important is using K-means clustering before the byte pair encoding?

Could use simpler disrectisation approaches (e.g. like the one used in the continuous control experiments here: https://arxiv.org/abs/2107.05431)

They mention filtering demonstrations

**Limitations:**

The paper does not have an explicit limitations section, which I think it could benefit from. For example, I would like to see a discussion of how this approach compares to settings where there is a very large amount of data to train offline RL agents. How does it compare then? In general, I would like to understand when this approach is the most suitable given data and simulator constraints.

I would also like to understand a little better how far this approach can go. For example, are there tasks at which point the vocab becomes filled very quickly and this approach is not applicable?

---

> ### Author Rebuttal · Authors · 2024-08-07
>
> Thank you for your time and care in reviewing our paper. We appreciate your feedback and share your excitement regarding the intuitiveness of the method and the computational efficiency. We answer your questions below.
>
> 1. How many demonstrations are used?
>
> For these tasks we use the existing D4RL dataset, and we give details on demonstrations in Appendix A. As to the quantity, we use ~1000 demonstrations for the AntMaze results, ~600 for the Kitchen results and ~100 for CoinRun. These numbers are defined (except in the case of CoinRun which we collect) by practices in prior work that we compare against. In Figure 6 we explore subsampling.
>
> 2. Can offline methods be used?
>
> D4RL was a dataset collected for offline RL, but the goal of our work is orthogonal. Even though offline RL might be used for these tasks, it assumes access to the observations, and the reward labels, whereas we only require the action sequences. In addition, because offline RL requires access to this extra information, it will memorize the particular layout of the scene in order to accomplish the task, and if the layout is rearranged, as in the transfer experiment, then new demonstrations and reward will have to be collected in the new layout. Thus, the use of offline RL is predicated on the availability of in-domain data, which is the opposite of exploration.
>
> 3. Are baselines SOTA?
>
> The baselines used are strong methods for unconditional learning, though the benchmarks tested in the literature are rarely shared. Our goal in this work is not only to compete for SOTA, but additionally to point out that a relatively simple and inexpensive method can compete with very expensive neural-network based solutions. In particula,  these NN-based solutions are so expensive that even running experiments takes substantially longer. The baselines are chosen to represent two common classes seen in the literature: models that generate subsequences with VAEs (SSP) and sequence models as priors (SFP).
>
> 4. How important is $k$-means before BPE?
>
> Without some kind of discretization step it would be impossible to run BPE as we need to merge the most common pair among a discrete set. k-means was chosen as it is a very standard way to discretize.
>
> 5. Could we use simpler discretization through binning?
>
> We did think about this possibility, but binning would result in an exponentially large number of discrete units (with the action space). This means that there could be a very large number of subwords discovered, which would result in a necessarily large final vocabulary. Such a large vocabulary makes RL difficult as we see in the paper. In addition, most of these discrete units discovered might not actually be necessary. In the example of the Ant, every possible motion of every leg joint is not necessary to make the Ant move, just the coordinated motions of lifting each leg up and down as a whole.
>
> 6. Missing limitations.
>
> We do address limitations in the conclusion section, but found it difficult space-wise to include a separate heading. We do not believe the comparison to offline RL is applicable as it is an orthogonal problem with different aims. Our goal is to improve exploration even when in-domain task data is not already available.
>
> 7. When is this method not applicable?
>
> Thanks for the question. We discuss this in the Conclusion section where we mention several limitations. In particular we speak about discretization removing resolution, which might be necessary for very fine-grained motor skills, and the open-loop execution, which would have disadvantages in stochastic environments. We believe some of these issues are addressable in the future.

---

> > ### Author Response · Authors · 2024-08-13
> >
> > We wanted to reach out to see if you had any remaining questions given that the discussion period wraps up in a few hours.
> >
> > Thank you again for your positive feedback.

---

> > > ### Comment · Area_Chair_6fx3 · 2024-08-14
> > > **Reviewer g8NK**
> > >
> > > Can the reviewer kindly engage with the authors?

---

### Official Review · Reviewer_iKv2 · 2024-07-12

**Soundness:** 2
**Presentation:** 2
**Contribution:** 3
**Rating:** 4
**Confidence:** 4

**Summary:**

This paper presents a novel method for skill discovery in reinforcement learning by leveraging tokenization techniques from Natural Language Processing (NLP). The approach involves discretizing the action space through clustering, and then using byte-pair encoding to generate temporally extended actions. The method is evaluated on various environments including AntMaze, Kitchen, and CoinRun, demonstrating improved performance over some state-free skill learning baselines and vanilla SAC.

**Strengths:**

This paper has several strengths:

- The perspective of using  NLP tokenization techniques for skill discovery in RL is creative and innovative.
- The method outperforms baselines in several sparse-reward environments.
- The study examines various aspects beyond performance, including computational efficiency, exploration behavior, and domain generalization.

**Weaknesses:**

While the paper demonstrates several strengths, there are also potential limitations and areas for improvement:

- The range of baseline skill discovery methods could be expanded. Additional algorithms, such as those proposed in [1] and [2], merit consideration.

- The presentation requires refinement. The current format of tables and figures impedes clear interpretation, potentially obscuring the full efficacy of the proposed method.

- The effectiveness of the skill discovery process appears to be heavily contingent on the specific dataset utilized. A more comprehensive exploration of dataset variability and its impact on outcomes would strengthen the study.

- There is no formal guarantee that the discovered skills are sufficient to construct an optimal policy. This theoretical limitation warrants acknowledgment and discussion.

- The computational complexity of the merging process may prove prohibitive for high-dimensional input domains, potentially limiting the method's scalability.

- It is unclear how the method would perform with larger, more diverse datasets that might require a larger skill space. This leaves questions about the approach's generalizability unanswered.

[1]  Y. Jiang, E. Z. Liu, B. Eysenbach, Z. Kolter, and C. Finn, “Learning options via compression,” arXiv preprint arXiv:2212.04590, 2022.

[2] A. Singh, H. Liu, G. Zhou, A. Yu, N. Rhinehart, and S. Levine, “Parrot: Data-driven behavioral priors for reinforcement learning,” arXiv preprint arXiv:2011.10024, 2020.

**Questions:**

See the "Weaknesses" section.

**Limitations:**

The authors have discussed a few limitations in the paper.

---

> ### Author Rebuttal · Authors · 2024-08-07
>
> Thank you for taking the time to review our paper. We appreciate that you highlighted the creativity of the approach as well as the breadth of the study. We respond point by point to your concerns below.
>
> 1. Range of baselines
>
> We would be happy to cite and discuss these methods in the paper. LOVE [1] is similar in its aims to our method, compressing sequences into reusable parts, but requires observation conditioning, which makes it hard to directly compare to our method. As discussed in Section 4.3, observation conditioning requires in-domain data collection and has tradeoffs with exploration. Parrot [2] is very similar to SSP or OPAL in that it is a flow model over actions, but instead of being over chunks of actions it is only over a single action at a time, so it would be as inefficient to run in practice as SFP, thus we thought SSP would be a better choice. The code is also not available.
>
> 2. Presentation requires refinement.
>
> It would be helpful to understand where precisely the issues are with the presentation so that we may amend them.
>
> 3. Dataset-specificity of skills
>
> To test the specificity of the dataset required to generate skills we provided experiments in the Hopper task in Appendix E, where we found indeed that quality was not so important, and we also subsampled the data in Figure 6, which suggests that only a very small number of decent demonstrations are necessary. If this is referring to the choice of tasks, we already choose more challenging tasks than prior methods.
>
> 4. No formal guarantee of optimality
>
> We agree that there is no formal guarantee of optimality and many skill-learning papers do not provide any such guarantees, including the baselines we consider. We already mention this limitation in lines 131-136, but we would be happy to expand this discussion if it is not sufficient.
>
> 5. Computational complexity of the merging process in high-dim
>
> Perhaps the presentation was unclear. In particular our merging process first consists of running k-means on actions and assigning them to their closest cluster centers. Then, we run BPE on the derived "strings". As such, the computational complexity of the merging process in high dimensions will only affect the k-means step, which indeed is more expensive for high-dimensional input, but still very very efficient on existing hardware for 1000-dimensional input spaces. The BPE step is on discrete units, so it will be fast regardless of the input dimension, certainly when compared to training neural networks. We hope this clarifies.
>
> 6. Unclear how method performs with larger, more diverse datasets
>
> RL with a large action space is difficult, but so is RL with an unconditional skill space that can perform many different behaviors. As such we don't believe this is any more difficult with our method than existing methods. We believe such studies are out of scope given the current large batch of experiments, but agree it would be interesting to follow up. Certainly our method would be much faster to test on larger datasets when compared to prior work given the speedups.

---

> > ### Comment · Reviewer_iKv2 · 2024-08-10
> >
> > Thanks for the response. I will maintain my original score.

---

> > > ### Author Response · Authors · 2024-08-11
> > >
> > > We made a concerted effort to address each of the issues that the reviewer raised and, while we respect the reviewer’s opinion, we would appreciate a justification for the decision. In our rebuttal we address all concerns and, with all due respect, we find a number of them to be broad or vague.
> > >
> > > As an example, when the reviewer says “the presentation requires refinement” without details as to specific changes desired, it is in direct contradiction to all other reviewers who praised the clarity of the presentation, thus it would be helpful to make specific comments.
> > >
> > > In addition, the comment that “computational complexity... may prove prohibitive in high dimensional input domains” is factually inaccurate. The current method scales quite nicely to high-dimensional action spaces as it only relies on K-Means, which has minibatch approximations in high dimensions.
> > >
> > > We already make these points in our rebuttal above, along with addressing other concerns. We understand that the reviewer has other commitments, but we would appreciate if they took the time to provide substantive feedback and a justification for their decision.

---

> > > > ### Author Response · Authors · 2024-08-13
> > > >
> > > > A friendly bump as there are only a few hours left in the discussion period.

---

### Official Review · Reviewer_bwJh · 2024-07-12

**Soundness:** 3
**Presentation:** 2
**Contribution:** 3
**Rating:** 6
**Confidence:** 3

**Summary:**

The paper titled "Subwords as Skills: Tokenization for Sparse-Reward Reinforcement Learning" introduces a novel method for skill extraction from demonstrations to address the challenge of exploration in sparse-reward reinforcement learning (RL). Inspired by the Byte-Pair Encoding (BPE) algorithm used in natural language processing, the authors propose a method to generate skills that can be used to accelerate both skill extraction and policy inference in RL tasks. The proposed method demonstrates strong performance in various tasks, providing up to 1000× acceleration in skill extraction and 100× acceleration in policy inference. The method also shows potential for skill transfer across loosely related tasks and generates a finite set of interpretable behaviors.

**Strengths:**

- **Originality**

The paper presents a novel approach by adapting the Byte-Pair Encoding (BPE) algorithm from natural language processing to skill extraction in reinforcement learning. This creative application demonstrates originality.


- **Quality**

The method is evaluated on several challenging sparse-reward RL tasks, such as AntMaze and Kitchen environments. The results indicate significant improvements in performance and efficiency compared to existing methods.


- **Clarity**

The paper is well-organized and clearly explains the methodology, experiments, and results. Visualizations and detailed explanations help in understanding the process and the outcomes of the proposed approach.


- **Significance**

The proposed method addresses a critical problem in RL, namely extracting skills from the demonstrations to solve exploration in sparse-reward environments. By significantly accelerating skill extraction and policy inference, the method has the potential to impact a wide range of applications in RL.

**Weaknesses:**

1. Stochasticity of the Environment

The impact of the environment stochasticity raises concerns. Since the method treats sequences of low-level actions as high-level actions, the cumulative effect of each low-level action can vary significantly in highly stochastic environments. For instance, if there is a 50% chance of wind blowing the agent off course, the resulting states from executing a single high-level action could vary greatly, potentially harming the algorithm's performance. The current evaluation environments, such as AntMaze and Kitchen, seem to have relatively low stochasticity. To strengthen the paper, it would be beneficial to include evaluation results from highly stochastic or procedurally generated environments, such as DMLAB30.

2. Intuitive Explanation of BPE Advantage

The paper would benefit from more intuitive explanations regarding why skills discovered using Byte-Pair Encoding (BPE) can outperform those discovered through simple k-means clustering. While BPE is known for its hierarchical and incremental construction of subwords in language processing, clarifying how these characteristics translate to improved skill discovery in reinforcement learning would enhance the understanding of the proposed method's advantages.

**Questions:**

1. Effect of Stochasticity on Algorithm Performance

How do you think the about stochasticity of the environment affects the algorithm's performance? Given the potential variability in outcomes when low-level actions are aggregated into high-level actions, understanding this impact is crucial.

2. Intuition Behind BPE Benefits

Can you provide some intuition on how BPE can be beneficial for skill discovery in reinforcement learning? Understanding the specific advantages of BPE over simpler methods of forming high-level actions like k-means clustering would help clarify the strengths of the proposed approach.

**Limitations:**

The authors acknowledge several limitations of their work:

1. **Resolution Loss Due to Discretization**: The discretization step in the skill extraction process reduces the resolution of the action space, which might not be suitable for tasks requiring fine-grained actions.
2. **Open-Loop Execution**: The method involves open-loop execution of subwords, which can lead to inefficient and unsafe exploration in certain scenarios. This limitation highlights the need for incorporating feedback mechanisms in future iterations of the method.

The paper does discuss these limitations openly.

---

> ### Author Rebuttal · Authors · 2024-08-07
>
> Thank you for your time and consideration spent reviewing. We appreciate that you highlight the novelty and significance of the approach, as well as the clarity in presentation and the challenge of the tasks that we choose to evaluate in. We respond to the weaknesses and questions below.
>
> 1. Stochasticity of the environment
>
> It is indeed the case that stochastic environments can pose issues, though we would like to clarify how exactly. The skills that we discover are 10 steps long, while the tasks that we set out to accomplish are hundreds of steps long. This means that skills tend to correspond to short sequences (e.g., taking a step or two forward, turning slightly, reaching toward an object), but not completely memorized actions. Thus if the environment layout were to change (as is the case in DMLab30, and in the case of our transfer experiments), it shouldn't pose an issue for our method as long as the policy can learn a generalized map from observation to action (which would be the case if many training environments were given). However, low-level stochastic dynamics would be more problematic and we discuss this in the limitations. We would be happy to expand this discussion. The deterministic environments we test are standard in the literature (including the baselines we use), so we did not consider this to be a deal-breaker. One option to mitigate this issue is to tune a residual policy on top of skills once they are discovered (not dissimilar to the residual correction in Behavior Transformers), though we find this to be out of scope for the current submission.
>
> 2. Advantage of BPE over simpler methods
>
> We believe there may be a slight miscommunication happening here, so we apologize for any lack of clarity in the writing. In short, we do not believe it is correct to compare "simple k-means clustering" with BPE for skill discovery as the two have entirely different purposes, and we use the first to seed the second. In particular, we want to discover high-level action *sequences*. Thus in order to use k-means clustering, we will need to define a distance metric over *sequences* of actions, which is tricky to do. One might think to average l2 distance over pairs of actions in different sequences, but if two sequences of scalar actions are shifts of each other, e.g. (a_1, a_2, a_3) =  (0, 1, 0) or (1, 0, 1) (i.e., alternating 0 and 1 actions), these would be farther apart than (0, 1, 0) and (0.5, 0.5, 0.5) which are completely different sequences. Thus it becomes difficult to define a "simpler" method operating on sequences. As to why we choose BPE, BPE is the simplest and most common method employed currently in NLP to discover discrete subsequences. It finds the most common subsequences in a greedy fashion, which in demonstration data would correspond to common behaviors useful across many different scenarios. Our insight is to reuse this technique in RL to help in solving the exploration problem, as these reusable behaviors should correspond to reusable skills. As we show in the paper, this method is much cheaper and faster than using sequence models for skill discovery as prior work has done. We hope this addresses any issues in communication and would welcome further questions to help clarify the presentation.

---

> > ### Author Response · Authors · 2024-08-13
> >
> > Thank you again for your feedback.
> >
> > The discussion period ends in a few hours. We would appreciate it if you could share any additional comments/questions you may have in light of our rebuttal above so that we can respond to them.

---

> ### Comment · Reviewer_bwJh · 2024-08-13
>
> Thank you for your responses, and I apologize for the late reply. After reading your rebuttal, I decided to increase the rating from 5 to 6. However, I still believe that exploring a systematic approach to dealing with highly stochastic environments could be an interesting direction.

---

> > ### Author Response · Authors · 2024-08-13
> >
> > Thank you for your reply, we appreciate the engagement. We're also happy that our rebuttal was able to change your opinion of the paper for the better. We agree that stochastic environments are an interesting direction, and in particular the idea of learning a residual correction on top of the low-resolution actions is something we may explore in the future.

---

### Author Rebuttal · Authors · 2024-08-07

We appreciate the time and care all reviewers have taken in reading our paper and offering feedback, and thank them for their input.

We are particularly happy to see reviewers appreciate the novelty and creativity of the proposed method (bwJh, iKv2), the extreme efficiency of the approach (bwJh, iKv2, g8NK, jRfS), the breadth of the evaluation (iKv2, g8NK, jRfS) and the quality of the presentation (bwJh, g8NK, jRfS).

We address concerns with the current draft in individual rebuttals for each reviewer below.

---

### Decision · Program_Chairs · 2024-09-25

**Decision:**

Accept (poster)

**Comment:**

This paper tackles reinforcement learning (RL) through a skill-learning framework to overcome the issue of needing to learn from sparse rewards collected only after a long, coordinated sequence of actions. The paper's approach is inspired by Byte-Pair Encoding (BPE) from natural language processing (NLP) to extract skills from demonstrations for re-use in RL. The paper shows that skills can be extracted from a small subset of the demonstrations and transferred to a new task.

The reviewers felt that the paper was interesting and warranted consideration. The reviewers appreciate the paper's originality, quality, clarity, and significance. However, reviewers also had concerns about the drift that can occur in a stochastic environment when focusing only on high-level actions (one reviewer with this concern raised their score), limited baselines, need for an improved presentation, reliance upon a high-quality dataset, no guarantees, computational complexity, insufficient details, and concerns about scalability.

The authors did a good job in seeking to improve the reviewers' assessments.

Remaining concerns after the rebuttal included a point about the inability to confirm the sufficiency of discovered skills to form a good policy and that the analogy of BPE in NLP translated to RL might not be an exact fit due to an inability to express the space of continuous control policies in robotics. Reviewers felt it might have been better to show a greater variety of tasks performed on the same robot embodiment.  However, other reviewers disagreed, arguing the current results were sufficient, and multiple reviewers argued for acceptance.